# Organogenesis of Plant Tissues in Colchicine Allows Selecting in Field Trial Blueberry (*Vaccinium* spp. cv Duke) Clones with Commercial Potential

**Ricardo Hernández** [1,†], **Alan López** [1,†], **Bárbara Valenzuela** [1], **Vivian D'Afonseca** [1,2], **Aleydis Gomez** [1] and **Ariel D. Arencibia** [1,*]

1   Centre of Biotechnology in Natural Resources, Faculty of Agricultural and Forestry Sciences, University Catholic of Maule, Ave. San Miguel, Talca 3605, Chile; ricardo.hernandez@alu.ucm.cl (R.H.); alan.lopez@alu.ucm.cl (A.L.); barbara.valenzuela.01@alu.ucm.cl (B.V.); vdafonseca@ucm.cl (V.D.); aleydisgomez1965@gmail.com (A.G.)
2   Pre-Clinical Sciences Department, Faculty of Medicine, University Catholic of Maule, Ave. San Miguel, Talca 3605, Chile
*   Correspondence: aarencibia@ucm.cl
†   These authors contributed equally to this work.

**Abstract:** Plants' genetic improvement continues to be crucial for modern agriculture, while biotechnology can offer efficient tools that enhance the selection and recommendation processes of elite clones. This work established a suitable methodology for the regeneration of blueberry (*Vaccinium corymbsum*) plants in cultures with colchicine. This could be considered a basis for producing populations for the selection of clones following a genetic improvement program assisted by biotechnology. The factors studied were: (a) explant type (leaf discs; nodal segments); (b) colchicine concentration (0, 0.5, 1, and 2 mg/L); and (c) time of exposure to colchicine (1, 2, 3, 5, and 30 days). The basal medium McCown's Woody Plant (WP) supplemented with 2 mg/L 2iP and 1 mg/L BAP was used with the commercial genotype Duke as a model. A total of 1957 blueberry clones were produced in a medium with 1 mg/L colchicine, distributed at different exposure times. Flow cytometry analyses revealed the following patterns: single patterns for random samples of control plants (Duke donor) and some clones regenerated on colchicine; double patterns for chlorotic plants regenerated on colchicine. Triple and quadruple patterns were observed in callus tissues that did not regenerate plants on colchicine. Populations of plants regenerated in colchicine (6787) and control plants regenerated in in vitro culture without colchicine were adapted under greenhouse conditions. The variables evaluated at this stage were adaptability, height, diameter, number of leaves, incidence of diseases, flowering capacity, and agrobotanical traits. Selected clones demonstrating phenotypic variability (157 clones) were transplanted to field conditions. From the clonal field trial conducted under minimum tillage conditions, 38 clones were selected for improved traits related to the agricultural yield and nutritional quality of the fruits. Of these, six clones showed the highest agronomic performance and adaptability to adverse environmental conditions compared to the Duke donor genotype. It is recommended that these clones continue genotype × environment interaction trials at different locations.

**Keywords:** highbush blueberry; plant improvement; biotechnology assisted genetic

## 1. Introduction

Southern highbush blueberry (SHB) types were originally developed in the 1980s by incorporating genes from species native to the southern United States to reduce the chilling requirement of new commercial varieties. SHB was first established in Florida and Georgia (1980s), then moved to north-central Chile (1980s), Argentina and Spain (1990s), and California (2000s), and, more recently, to Mexico, Peru, and Ecuador (2010) [1].

Within the genus *Vaccinium*, which includes approximately 400 species, blueberry comprises between 10 and 26 of them, according to its taxonomic classification [2]. Highbush blueberries, which account for the largest area of commercial cultivation worldwide, are subdivided according to their winter chilling requirements into northern and southern types [3]. At a commercial level, blueberries are grown in diverse geographical areas with temperatures and solar radiation that differ significantly from those of their origin; therefore, the selection of new varieties with higher adaptability is a key issue [4].

The world import market for fresh fruit from Mediterranean climates has increased by 7.3% in volume and 34.6% in value. The highest volume growth in this market is observed in blueberries (*Vaccinium corymbosum* L.) at 44% [5].

Previous work [6] has already addressed the challenges for the micropropagation of blueberries with commercial interest in temporary immersion bioreactors (TIBs) that maintain the genetic uniformity of the genotypes; however, no programs have been established to broaden the genetic base for blueberries in Chile. Plant genetic improvement can also be assisted by biotechnological approaches, among which the efficiency of traditional breeding can be improved using marker-assisted selection (MAS), but it remains a challenge for crops such as highbush blueberry, which is characterized by polyploidy and a highly heterozygous genome [7]. One of the most widely used procedures to improve these biotic and abiotic traits is to apply colchicine treatments to induce different degrees of genetic variability resulting from polyploidy [8–10]. Highbush blueberry breeding mainly focuses on low chilling requirements, superior fruit quality [11], increased shelf life, and tolerance/resistance to major pests and diseases. Conventional breeding through germplasm selection or intra- and interspecific hybridization has been widely used for the development of new highbush blueberry varieties [11–13].

Tissue culture technology has proven to be a flexible platform that supports plant breeding programs. The genetic improvement of this species has also been achieved through genetic engineering [14–17]. For example, through the integration of genetic engineering, mutagenesis, or somaclonal variation in in vitro culture it is possible to express, induce, or evidence genetic variability as a source for the selection of elite clones for plant breeding [18].

Colchicine is an alkaloid metabolite obtained from meadow saffron (*Colchicum autumnale* L.) that inhibits chromosome separation during cell division, leading to chromosome duplication, and it is used as an antimitotic agent by binding to tubulin dimers, preventing microtubule formation during cell division [19]. It is widely used as an antimitotic agent for polyploidy induction [20–23]. At optimized concentrations, they allow for the replication of the plant genome, but, in cases in which cell division does not occur, there is a probability of variation at the somaclonal level. Polyploidy is considered an important breeding method because polyploid organisms have higher vigor and, in some cases, are superior to diploid organisms of the same species in traits such as higher yield, better product quality, and increased tolerance to biotic and abiotic stresses [22]. Polyploidy in the form of tetraploids can be associated with major evolutionary transitions, large developmental leaps, and/or adaptive species radiation [23]. Quantitative (polygenic) traits, whose phenotypes are the result of both gene action and environmental influences, are often referred to as multifactorial or complex traits. Many polyploidy induction processes have been reported with colchicine, which is an effective mechanism; however, it has also been observed that the use of colchicine generates chimerization and death of the tissue with which it has been in contact, so that the production of viable polyploid plants is diminished in a large proportion [24,25].

In this study, blueberry (*Vaccinium corymbosum* L. cv Duke) explants were induced in media with different colchicine treatments. In vitro regenerated clones and non-regenerable calli were analyzed using flow cytometry. Blueberry clone populations regenerated in colchicine were adapted in greenhouses, evaluating the phenotypic variability of agrobotanical characters. Clones selected for their vigor, adaptability, and flowering were evaluated for agroproductive traits under field conditions. The objective of this study was to establish a technological platform for the regeneration of blueberry plants (*Vaccinium corymbosum* L.)

with genetic variability, followed by a selection scheme that enables biotechnology-assisted genetic improvement of commercial cultivars in the Maule Region, Chile.

## 2. Materials and Methods

### 2.1. Plant Materials

Blueberry seedling stocks of the genotype Duke were established in vitro, as follows. Adventitious stems of the donor plant (~5 cm) were surface disinfected by treatment in 70% ethanol (5 min) followed by a commercial solution of 10% NaOCl and 0.1% Tween 20 (15 min) and then rinsed three times in sterile distilled water [26]. Shoot tips were then grown for 8 weeks in McCown's Woody Plant (WP) medium supplemented with 30 g/L sucrose and 1 mg/L 2iP, as previously described in other works [27,28]. The culture medium was solidified with 6 g/L agar, and the pH was adjusted to 5.2 before autoclaving at 121 °C for 20 min. Flasks of plant cultures were maintained at 23 $\pm$ 2 °C for a 16/8 h photoperiod under a combination of natural light and cool white fluorescent tubes at a light intensity of 60 $\mu$M m$^{-2}$s$^{-1}$. For in vitro micropropagation, nodal segments of in vitro proliferated explants were subcultured monthly in culture medium, as described above.

### 2.2. Colchicine Treatments

From 21-day-old blueberry vitroplants, two types of explants were tested for plant regeneration: a- leaf discs from the upper third; b- trinodal segments. To determine the regeneration efficiency of blueberries in a medium with colchicine, a bifactorial experiment was carried out considering the types of explants previously described, combined with the following concentrations of colchicine: 0 (control), 0.5, 1, and 2 mg/L. The experiments were replicated three times for 21 days in each replicate. In the second validation stage, an explant/colchicine concentration combination was selected and combined with different treatments (i.e., exposure times) to colchicine: 1, 2, 3, 5, and 30 days. A total of 1000 explants per treatment were studied. Each regenerated individual shoot was considered an independent clone and micropropagated in 1 mg/L colchicine for an additional 30 days.

### 2.3. Flow Cytometry

The variation in DNA content was determined by flow cytometry after treatment with colchicine and in plants micropropagated without colchicine (i.e., control). For the analysis, the following blueberry materials were randomly selected: 10 green regenerated plants (control without colchicine), 10 regenerated plants treated with colchicine (different intensities of green color, chlorotic), and 10 non-regenerable callus tissues induced by colchicine. The samples were cut into small pieces in a Petri dish, and 200 $\mu$L of solution A from the Plant High Resolution DNA kit type P (Partec) was added to break the membrane and isolate the nuclei; in addition, 1 mL of DAPI staining solution [10 mM Tris-HCl, pH 7.5, containing 50 mM sodium citrate, 2 mM MgCl2, 1% (*w/v*) PVP K-30 (Wako Pure Chemicals Industry Ltd., Osaka, Japan), 0.1% (*v/v*) Triton X-100, and 2.5 mg/L DAPI (4',6-diamidino-2-phenylindole dihydrochloride)] was added to stain the nuclei. After incubation for a few minutes, the mixture was filtered and subjected to flow cytometry analysis using a PA flow cytometer (Partec GmbH, Munster, Germany) equipped with a Hg lamp.

### 2.4. Acclimatization in Greenhouse Conditions

Each clone was carefully separated, washed in water, and planted in 128-cell plug trays (cell volume 125 cm$^3$) containing a mixture of pine compost and zeolite (2:1). The trays were kept in a greenhouse, and the relative humidity was gradually reduced (90% $\rightarrow$ 80% $\rightarrow$ 70%) at 10-day intervals under a natural photoperiod. Clones that survived transplanting under greenhouse conditions were evaluated for two years for the following agrobotanical traits: plant vigor, leaf shape, axillary bud distribution, leaf coloring, and flowering capacity.

### 2.5. Field Trial for Selection of Elite Clone

The selected clones (157) were transplanted to field conditions in an experimental plot in the locality of San Antonio de Encina ($-35.85902$, $-71.49819$), Linares Province. For three years, the experimental plot (0.5 ha) was maintained under conditions of minimal soil tillage, without pruning or plantation management, as well as without applications of chemical or biological products. Crop morphological traits under field conditions were measured quantitatively with a caliper to measure main stem thickness (basal and apical) and plant height and to determine the intercrop variability, similar to that described in other works [29,30]. The presence or absence of pests and disease symptoms was assessed qualitatively in the clones under natural conditions. In the 2023–2024 season, the following agro-productive variables were quantitatively evaluated: harvest per plant (g); number of fruits per plant; weight, diameter, height, and elliptical volume of fruits (10 fruits for 3 replicates); maturity Brix was measured with a hand-held refractometer for field conditions (0–30° range) and density of juices.

### 2.6. Characteristic Dispersion and Variability

For a comparative study among traits, a normalization algorithm (0–1) was designed for the data sampled and collected in Excel. This was performed in Python code with the software JupyterLab (3.0.14), in which the dispersions and correlations among the measured variables were plotted using the matplotlib library.

### 2.7. Statistical Analysis

After checking the normality of the data using the Kolmogorov–Smirnov test and the homogeneity of the variance with Levene's statistic, which were significant in both cases ($p < 0.05$), the Kruskal–Wallis (nonparametric ANOVA) and Dunn's tests were applied for multiple comparison. Comparisons among groups were established according to colchicine exposure times (i.e., treatments). These analyses were performed using the IBM SPSS Statistics V25 software.

Principal component and cluster analysis (PCA) was performed in RStudio 2023.09.0 + 463 using the libraries factoextra, ggplot2, readxl, and cluster. Briefly, the FactoMineR package in R was applied to reduce the dimensionality of the data and explore the inherent patterns. We visualized the results using graphs representing the distribution of observations and contribution of the variables to the principal components. In addition, we implemented a k-means algorithm to identify natural clustering in the data, facilitating the identification of possible patterns or categories of blueberry performance. To determine the optimal number of clusters, we used an algorithm based on the silhouette method that assesses the cohesion and separation among clusters.

## 3. Results

### 3.1. Blueberries Production in Colchicine

In this study, the regeneration of *V. corymbosum* L. proliferated explants in colchicine treatment was developed. In this way, the regeneration efficiency of two types of explants, leaf discs and nodal segments, in culture media with different concentrations of colchicine, was compared. The results at 21 days showed the highest plant regeneration values in the control treatments (without colchicine), highlighting the nodal segments (with three buds) with 82% regeneration efficiency, whereas the leaf discs were only 47% efficient (Table 1; Figure 1A–C). As expected, colchicine decreased the efficiency of plant regeneration, being completely suppressed at a concentration of 2 mg/L in both types of explants; this treatment generated only 8% of mucilaginous calluses in the explants of nodal segments, which did not regenerate whole plants.

**Table 1.** Summary of regeneration rate of blueberries in different concentrations of colchicine in leaf discs and nodal segments isolated from Duke vitroplants.

| Explants | Colchicine | | | | | | | |
|---|---|---|---|---|---|---|---|---|
| | 0 mg/L (Control) | | 0.5 mg/L | | 1 mg/L | | 2 mg/L | |
| | VS | NRC | VS | NRC | VS | NRC | VS | NRC |
| Leaf discs * (upper third) | 47 (47%) | 0 | 21 (21%) | 9 (9%) | 11 (11%) | 16 (16%) | 0 | 0 |
| Nodal segments ** (three buds) | 246 (82%) | 0 | 94 (31%) | 25 (8%) | 36 (12%) | 42 (14%) | 0 | 24 (8%) |

\* Total of 100 leaves per treatment. \*\* Each explant (nodal segment) has three potentially regenerable buds. VS: viable shoots; NRC: non-regenerable calli.

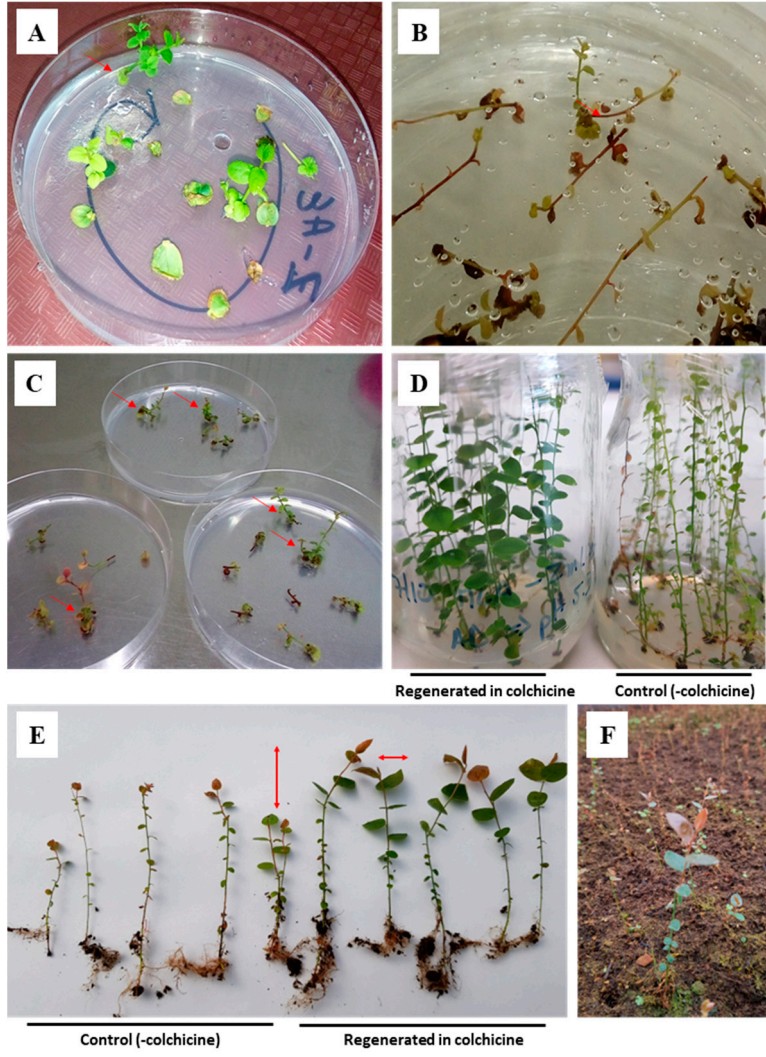

**Figure 1.** Regeneration of blueberry plantlets from different explants in a medium supplemented with 1 mg/L colchicine: (**A**) regeneration from leaf discs; (**B**,**C**) regeneration from nodal segments; (**D**) blueberry multiplication (clone) in 1 mg/L colchicine (left) and control without colchicine (right); (**E**) rooting of blueberry clones micropropagated in 1 mg/L colchicine, comparison of vigor between control plants (left) and regenerated plants in the colchicine treatment (right); (**F**) colchicine-treated clone grown under greenhouse conditions.

Treatments with 0.5 mg/L and 1 mg/L colchicine resulted in a decrease in regeneration and an increase in the production of non-regenerable calluses, which could be explained by

the effect of colchicine on the organogenesis process. With the criterion of a balance between the regeneration capacity of the plants and the probability of occurrence of genomic changes due to the interaction with colchicine, a combination of nodal segment explants in medium with 1 mg/L of colchicine was selected for subsequent experiments.

To achieve populations of blueberries regenerated in colchicine, scaling experiments were conducted with exposure times of nodal segments to colchicine of 1, 2, 3, 5, and 30 d. A summary of these experiments is provided in Table 2, which shows that treatment without colchicine (i.e., control) generated more than three shoots/explants, corroborating the high efficiency of the regeneration medium used. In general, the results showed the effect of colchicine in the treatments studied, with a notable decrease in the variables plant regeneration capacity (number of initial clones) and adaptation efficiency of vitroplants to greenhouse conditions. The efficiency of the initial plant regeneration ranged from approximately 8 to 16%, while the adaptation to the greenhouse was between 54 and 60%. On the other hand, the variable multiplication rate in colchicine and the percentage of plants adapted to greenhouses had values in the range of 1:5–1:8 and 73–95%, respectively. Plants regenerated and subsequently micropropagated (i.e., cloned) in colchicine showed phenotypic variability with greater vigor, intense green leaf area, and profuse in vitro rooting (Figure 1D,E). This phenotypic variability was maintained in plants transplanted to greenhouses, demonstrating greater adaptability to environmental conditions (Figure 1F).

**Table 2.** Summary of experiments to generate blueberry populations regenerated in vitro from nodal segments at different treatment times with 1 mg/L colchicine.

| Treatments | Number of Explants * | Regenerated Shoots (Clone)/% | Percent | Multiplication in Colchicine | Rate | In Vitro Rooted Plants | Percent | Plants Adapted in Green-house | Percent ** | Clones Trans-planted to Field | Percent **** |
|---|---|---|---|---|---|---|---|---|---|---|---|
| 1 day | 1000 | 405/13.5 | 13.5 | 2041 | 1:5 | 1884 | 92.3 | 1149 | 60.9 | 15 | 3.7 |
| 2 days | 1000 | 484/16.1 | 16.1 | 3621 | 1:7 | 3427 | 94.6 | 1988 | 56.8 | 18 | 3.7 |
| 3 days | 1000 | 262/8.7 | 8.7 | 1560 | 1:6 | 1492 | 95.6 | 821 | 55 | 21 | 8 |
| 5 days | 1000 | 421/14 | 14 | 3455 | 1:8 | 3112 | 90.1 | 1891 | 60.8 | 49 | 11.6 |
| 30 days | 1000 | 385/12.8 | 12.8 | 2310 | 1:6 | 1708 | 73.9 | 938 | 54.8 | 54 | 14 |
| Total | 5000 | 1957/39.14 | | 12 987 | | 11 623 | | 6 787 | | 157 | |
| Control (-colchicine) | 1000 | 3426/114.2 *** | 114.2 *** | N.C | N.C | 3912 | 114.1 *** | 3708 | 94.7 | - | - |

* Each explant (nodal segment) had three potentially regenerated buds. ** Plants adapted in greenhouse vs. plants rooted in vitro. *** Because of the effect of phytohormones, some buds/plants generated two individuals in vitro. **** Clones that show phenotypic variability and flowering (number of clones regenerated in vitro/number of clones selected for transplant into the field).

In summary, 1957 plants regenerated in colchicine (1 mg/L) were produced and coded as independent clones. These clones were micropropagated in a medium with colchicine (1 mg/L), generating a total of 12,987 plants; of these, 11,623 rooted in vitro, of which 6787 adapted to greenhouse conditions. It should be noted that even considering the losses at different stages, all clones were represented in the population studied in the greenhouse.

### 3.2. Flow Cytometry Analysis

A flow cytometry analysis was performed to verify the relative DNA content in regenerated plants and induced callus tissues in the colchicine (1 mg/L) medium. The following materials were randomly selected: 10 control plants regenerated in a medium without colchicine, 10 plants regenerated in colchicine, and 10 callus tissues induced in a medium with colchicine. Among the plants (clones) regenerated in colchicine, phenotypic variability in vigor was evidenced, and both green and vigorous plants were analyzed, as well as clones with opaque yellow–green coloration, with less vegetative development in vitro.

The results show a unique DNA pattern in the control plants regenerated in colchicine (Figure 2). This simple DNA pattern was also observed in colchicine-regenerated clones with high phenotypic vigor and green leaves. On the other hand, the double-DNA pattern

was observed only in some clones regenerated in colchicine, indicating that these were less vigorous plants with a grey yellowish-green color; these plants did not survive the rooting-adaptation period under greenhouse conditions. More complex patterns (three and four DNA peaks) were also determined; however, these were from non-regenerative calli with a non-friable and mucilaginous phenotype. In summary, no relationship was found between complex DNA patterns and phenotypic variability evidenced in vitro, putatively associated with polyploidy in plants regenerated in colchicine. The analyzed clones regenerated in colchicine, which showed high phenotypic vigor in vitro, showed a simple DNA pattern, similar to the control plants regenerated in medium without colchicine.

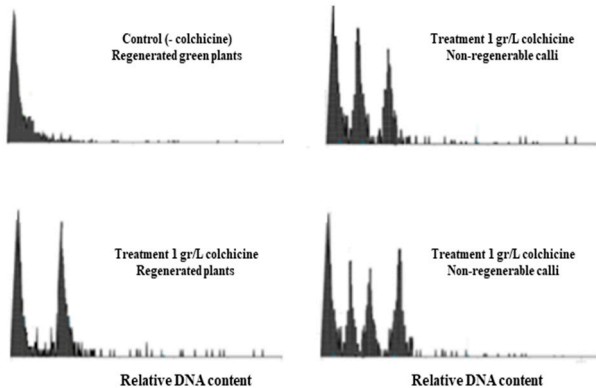

**Figure 2.** Flow cytometry patterns in in vitro plants and calluses of blueberries grown in medium supplemented with 1 mg/L colchicine.

### 3.3. Evaluation of Agrobotanical Traits in Greenhouse Conditions

To establish a bridge with traditional genetics, the strategy of evaluating the entire population of blueberry plants resulting from colchicine treatment was followed under environmental conditions. For this stage, the hypothesis that the genotype–environment interaction would allow for verifying the evidenced/induced genetic variability, and its stability in the regenerated blueberry population under the previously described conditions was considered.

A population of 11,623 in vitro rooted plants corresponding to 1957 clones (regeneration events in colchicine) were transplanted to greenhouse conditions in the locality of San Antonio de Encina, Maule Region, Chile ($-35,859$, $-71,498$). Of these, 6787 plants (58.4%) were adapted to greenhouse conditions. After three months of adaptation, genetic variability was observed among plants of the same clone for agrobotanical characteristics such as size and color of the leaves, for which these individuals were recoded (Figure 3A). The efficiency of the adaptation to the greenhouse varied among the clones (Figure 3B,C), while for some clones all the plants died.

The individuals adapted to greenhouses were evaluated for one year, demonstrating differences in agrobotanical characteristics, such as the shape and color of the leaves (Figure 4A), distribution of secondary branches on the stem (Figure 4B), leaf area and number, and distribution of leaves (Figure 4C). Leaf shapes varied from rounded to lanceolate or heart-shaped, with color gradations from purple–red to deep green. It was notable to find clones with many small secondary branches, small and firm leaves with phenotypes very distant from the donor genotype, some with a herbaceous appearance, and creeping growth, which were discarded because they did not have interesting traits for the genetic improvement of blueberries.

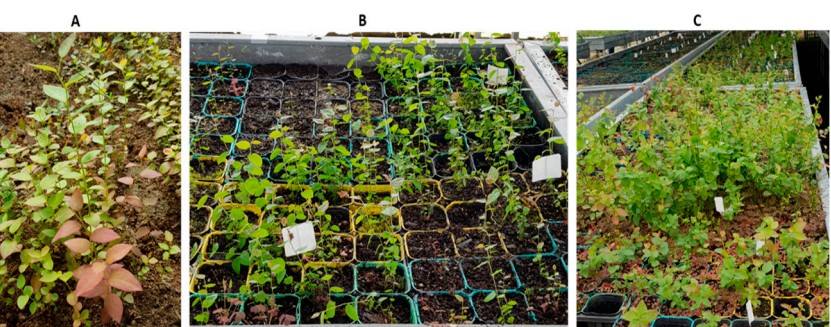

**Figure 3.** Greenhouse adaptability of blueberries grown in colchicine: (**A**) individual evidence of phenotypic variability between clones; (**B**,**C**) differences in adaptation efficiency between clones after three and six months.

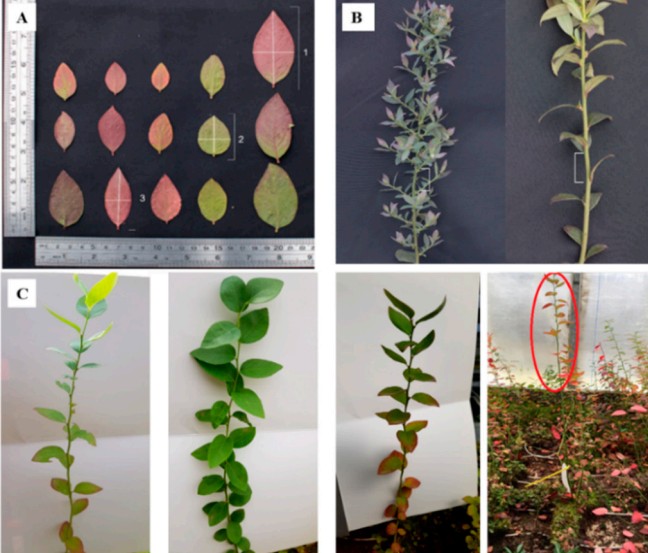

**Figure 4.** Examples of phenotypic variability among blueberry clones regenerated in colchicine: (**A**) shape and color of the leaves (each column corresponds to a clone); (**B**) distribution of secondary branches on the stem; (**C**) leaf area and number, distribution of leaves, and right-contrast (highlighted in red) in plant height of a selected individual.

In parallel, quantitative variables related to the vigor of the individuals were evaluated. Figure 5 shows the variability represented by the wide dispersion of the individuals in the plane (there are overlapping points) and not demonstrating a correlation between the studied variables and colchicine treatments. In the case of the plant height variable, the highest concentration of individuals was evident in the lower right area of the plane (Figure 5A), while for the variable leaf number, it was evident in the central left area of the plane (Figure 5B). Conversely, the stem diameter variable showed the highest concentration of individuals in the central part of the plane (Figure 5C), similar to the variable leaf area, although in this case there was a greater dispersion of the individuals (Figure 5B). It is important to highlight that, for all of the studied variables, the presence of contrasting individuals was demonstrated, so it is possible to verify the potential to select blueberry individuals with high vigor at this stage.

The results of the nonparametric Kruskal–Wallis analysis for variables related to vigor in blueberry clones regenerated at different exposure times (1, 2, 3, 5, and 30 days) to colchicine (1 mg/L) are shown in Figure 6. In the case of the variable stem basal diameter, a significant difference was demonstrated among the treatments, with the highest values being in the population of regenerated plants with an exposure time of 30 days in medium with colchicine (Figure 6A). For the variable stem apical diameter, the highest values were

observed in the treatments of 3 and 30 days of regeneration time in colchicine (Figure 6B), while for the variable plants height the peak values were also observed in clones regenerated in the treatment of 30 days (Figure 6C). In summary, it is verified that the clones regenerated in the longest time of exposure to colchicine (30 days) are related to plants with the highest vigor.

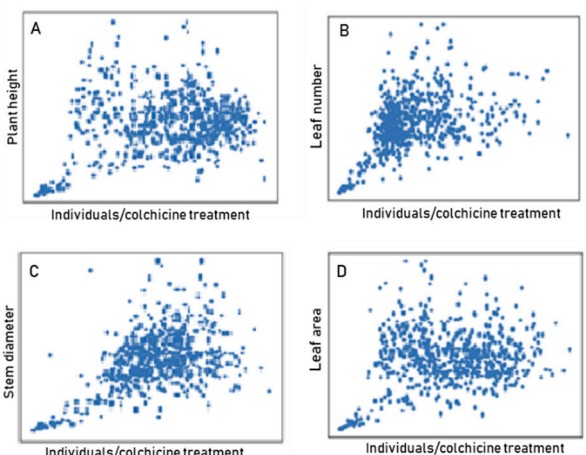

**Figure 5.** Genetic variability for variables related to vigor in populations of blueberry individuals regenerated in colchicine: (**A**) plant height; (**B**) leaf number; (**C**) stem diameter; (**D**) leaf area.

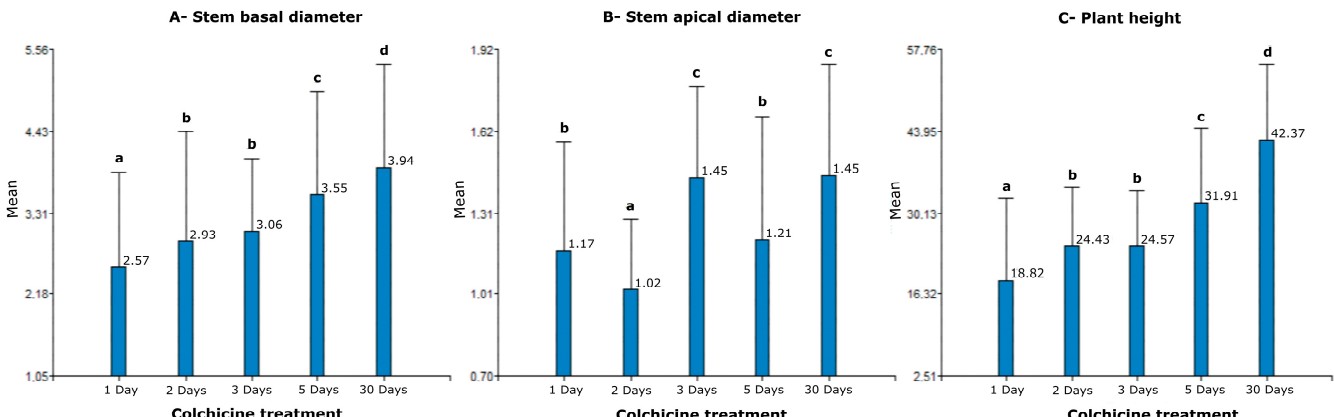

**Figure 6.** Non-parametric Kruskal–Wallis analysis for variables related to vigor in a population of blueberries regenerated in different treatments with colchicine. (**A**). Stem basal diameter; (**B**). Stem apical diameter; (**C**). Plant height. Different letters indicate significant differences ($p < 0.05$).

Although this increase in vigor increases among the treatments studied for the variables basal diameter of the stem and plant height, this does not seem to be proportional to the time of exposure to colchicine. For the stem apical diameter, the previous relationship is not evident, possibly explained by the variability in growth habit, a variable that in this study was considered qualitative.

In the second year of adaptation in greenhouses, the evaluation for flowering was carried out during the months of September–November, spring in the southern hemisphere. Figure 7 shows the results of the correlation between plant vigor (measured by height and stem diameter) and flowering (presence vs. absence) of blueberry populations regenerated in the different treatments with colchicine.

The results display the distribution of clones with and without the ability to flower under the studied conditions (Table 2), as well as the variability in the shape and color of the flowers, leaves, and stems that was maintained for a period of two years (Figure 7). In the case of the treatments of 1, 2, and 3 days of exposure to colchicine, a lower number

of flowering clones were observed in relation to those that did not flower. In contrast, in the clones from the 5 and 30 day colchicine treatments, a superior number of clones with the capacity to flower are present, being notable that they corresponded to plants with greater vigor relative to higher heights and diameter of the stems. This could be explained by the approach of a second stage of multiplication in medium with colchicine, which should potentially increase the effect on the tissues, in addition to controlling the possible appearance of mosaic plants considering that we based it on a regeneration system through organogenesis. It should be noted that in all colchicine treatments there were clones that were separate and appeared out of the group.

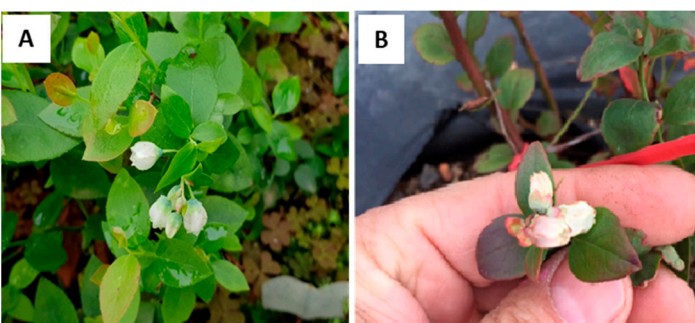

**Figure 7.** Genetic variability for flowering (2nd year) under greenhouse conditions: (**A**,**B**) examples of phenotypic variability among clones for agrobotanical traits related to flowers and leaves.

After transplanting from greenhouses and for three additional years, the field trial stage was carried out under minimum tillage, without agronomic management or pruning, dry conditions and without applications of chemical or biological products. The experimental plot is surrounded by blueberry fields in commercial exploitation. This strategy was followed to increase the selection pressure on the clonal population of blueberries from colchicine treatments. As a result, it was obtained that of 157 blueberry clones transplanted into the field, 38 demonstrated high plasticity and adaptability to the environmental and management stress conditions explained above (Figure 8). The selected clones showed greater vigor and intense green coloration in the leaves and branches. Additionally, the plants' health was considered as a selection criterion, indicated by the absence of symptoms of major disease and pest incidences. In contrast, the Duke donor clone (control) had greenish-yellowish leaves, with reddish leaf areas, the plant being less vigorous compared to the selected clones (regenerated in colchicine).

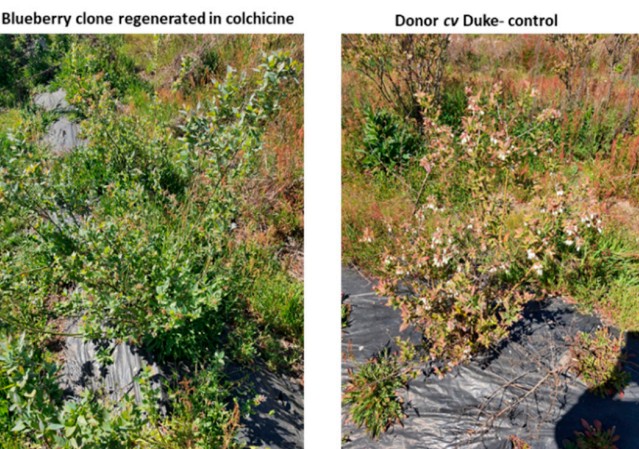

**Figure 8.** Genetic improvement for adaptability to conditions of minimal soil tillage and sustainable management. Note the differences in plant vigor and leaf color between a blueberry clone regenerated in colchicine (**left**) and the donor genotype Duke control (**right**).

The plant vigor also showed an influence on the fruiting variables related to agricultural yield, while differences were demonstrated between the selected clones and the Duke donor genotype (Figure 9). In this case, the values of the fruit yield variables (number of fruits/plant, weight, diameter, height, and ellipsoid volume) were lower for the Duke control (A) compared to the elite clones regenerated in colchicine (B; C; D).

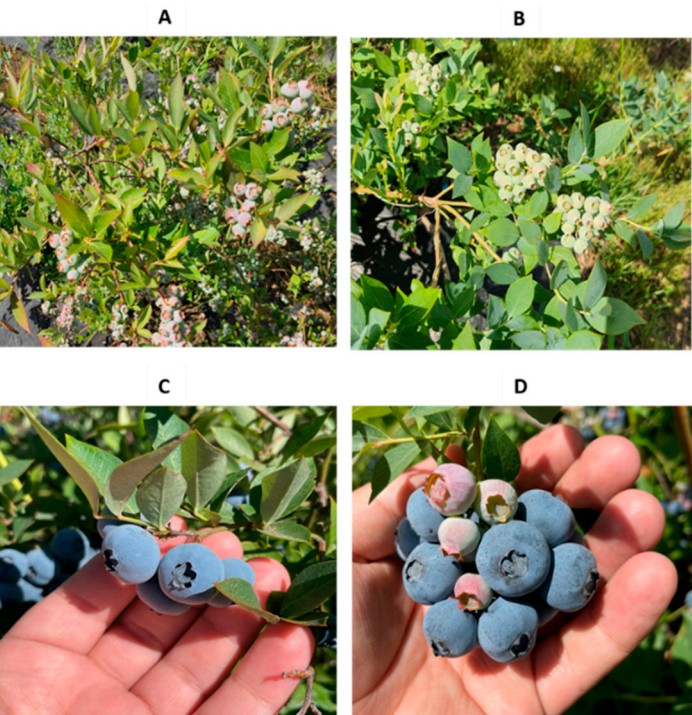

**Figure 9.** Genetic improvement in fruit production in blueberry clones after five years of being regenerated in colchicine: fruiting in the control donor cultivar Duke (**A**); in the elite clones 1–436 (**B**); ripe fruits in elite clones 18–420 (**C**); elite clones 1–436 (**D**).

In this case, the values of the fruit yield variables (number of fruits/plant, weight, diameter, height, and ellipsoid volume) were minor for the Duke control (A) compared to the elite clones regenerated in colchicine.

The multivariate analysis of the principal components (MCPA) shows that the first component (C1) contributed 59.1% of the experimental variance, while the second component (C2) contributed 29.2% Overall, C1 and C2 explain 85.3% of the total variability, validating the experimental design used (Figures 10 and 11).

In this sense, the variables that explain C1 are, in this case, those associated with the yield, in the following order: fruits size (volume, height, and diameter), fruits weight, gr of fruits (harvest) per plant, and number of fruits per plant. On the other hand, the variables that explain C2 are the brix and the density of the juices, both related to the fruit's nutritional quality (Figure 10).

Furthermore, the PCA consistently defines two groups of blueberry clones, classified as high yielding and low yielding in fruit production (Figure 11). While the Duke control genotype was grouped in the lowest yield, several clones were distributed around the center of gravity of the high yield C1, highlighting six clones with larger and weightier fruits, these being key variables of agricultural yield.

In the six clones selected as elites (Table 3), increases were determined with respect to the Duke donor for the following variables associated with fruit yield: weight ($\Delta$ range between 0.92 and 1.69 g/fruit); diameter ($\Delta$ range between 0.31 and 0.50 cm); height ($\Delta$ range between 0.2 and 0.32 cm); and ellipsoid volume ($\Delta$ range between 7.40 and 12.04 mm$^3$).

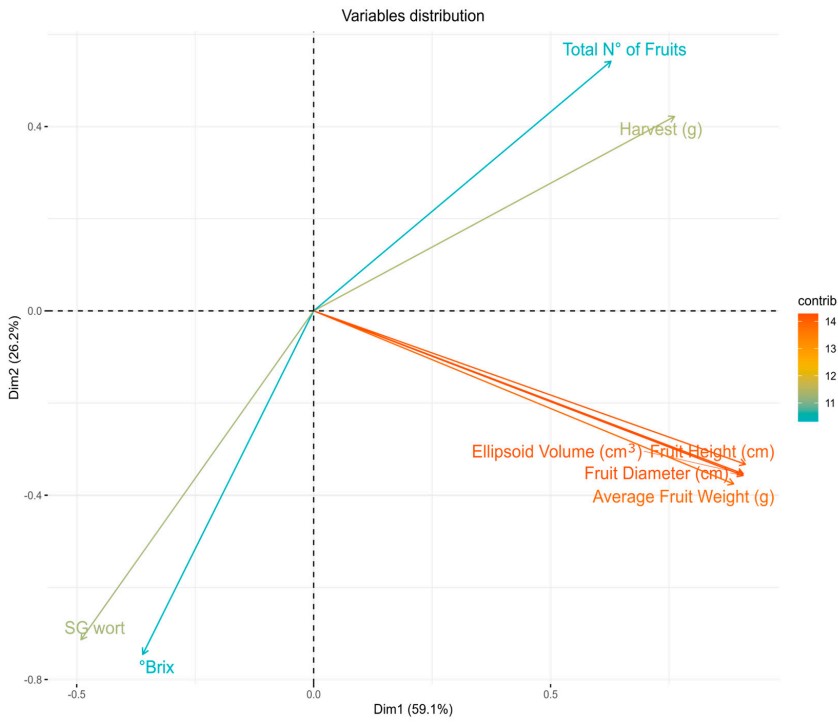

**Figure 10.** Principal component analysis (PCA): contribution of the variables associated with the agricultural yield and fruit quality in the population of blueberries regenerated in colchicine studied for the 2023–2024 harvest.

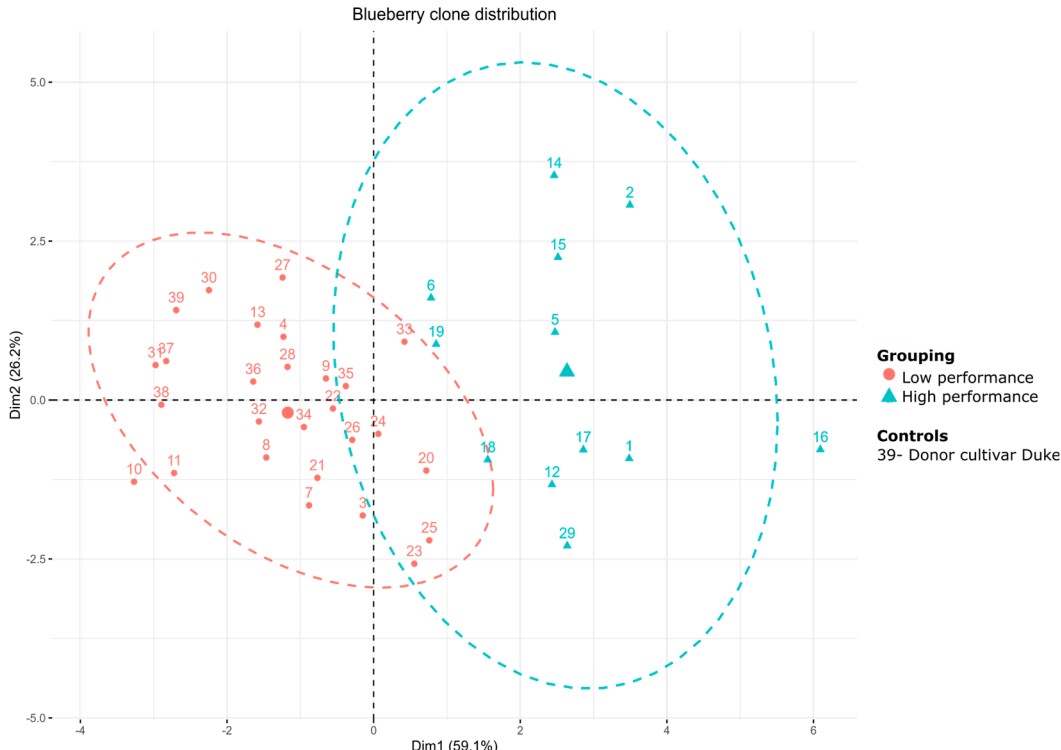

**Figure 11.** Cluster k-means analysis. The number of clusters was determined by the silhouette method. The components are the same as those obtained in the principal component analysis (PCA): grouping of elite clones (C1–2) in relation to the agricultural yield variables in the population of blueberries regenerated in colchicine determined via field trials for the 2023–2024 harvest.

**Table 3.** Genetic improvement related to components of agricultural yield in blueberry clones selected under minimal agronomic management and grouped as elites according to PCA statistics.

| Clone | Weight (g) * | Δ * | Diameter (cm) * | Δ * | Height (cm) * | Δ * | Ellipsoid Volume (cm³) * | Δ * |
|---|---|---|---|---|---|---|---|---|
| 1–436 | 3.11 b | 1.68 | 1.91 a | 0.5 | 1.35 b | 0.29 | 20.63 b | 11.8 |
| 5–437 | 2.35 c | 0.92 | 1.76 c | 0.35 | 1.26 c | 0.2 | 16.35 c | 7.52 |
| 12–102 | 2.92 c | 1.49 | 1.90 b | 0.49 | 1.34 c | 0.28 | 20.26 c | 11.43 |
| 17–421 | 2.84 c | 1.41 | 1.82 c | 0.41 | 1.33 c | 0.27 | 18.45 c | 9.62 |
| 18–420 | 2.67 c | 1.24 | 1.72 c | 0.31 | 1.31 c | 0.25 | 16.23 c | 7.40 |
| 29–440 | 3.12 a | 1.69 | 1.90 b | 0.49 | 1.38 a | 0.32 | 20.87 a | 12.04 |
| (39) Donor ** | 1.43 d | - | 1.41 d | - | 1.06 d | - | 8.83 d | - |

* Average of ten fruits per three replicates. ** Genotype Duke, Δ (increase compared to the donor). Different letters indicate significant differences ($p < 0.05$).

## 4. Discussion

This research reports the establishment of a biotechnology-assisted scheme for the genetic improvement of blueberry (*V. corymbosum* L.). In summary, from a population of plants regenerated in in vitro culture with colchicine, the stages of adaptation and selection were followed, which resulted after five years in the recommendation of six elite clones with improved traits referring to variables of agricultural yield and adaptability.

Firstly, regeneration via organogenesis of blueberry clones was standardized from nodal segments isolated from explants of the donor cultivar Duke at different colchicine exposures (1 mg/L). Subsequently, the adaptation of clones in the greenhouse was achieved, laying the basis for population trials under field conditions. In blueberries, it has been demonstrated that the most efficient explants to induce shoots and roots of *V. corymbosum* are the apex of shoots and explants of two nodal segments, which were isolated from the in vitro plants of the cultivars Farthing and Legacy [24]. In this same sense, in a study also using shoot apices and nodal segments of Legacy, the authors concluded that nodal segments were the best starting materials for the induction of shoots and blueberry seedlings [31].

Among the clones regenerated in colchicine, the presence of an early phenotypic variability in vitro was demonstrated (after 30–40 days of culture), essentially for traits related to the plants' vigor. This fact constitutes the first evidence of the potential of the established methodology for the production of blueberry clones for the further selection process in ex vitro environments. In this way, at the end of the in vitro culture stage in random samples of plants regenerated in colchicine that showed phenotypic variability, the nuclear DNA content was determined. In addition, samples of non-regenerable callus that were induced in medium with colchicine were included in the DNA determination.

According to our results, it was not possible to correlate the DNA patterns (i.e., peaks) and the increase in ploidy between in vitro plants regenerated in colchicine. In this sense, complex patterns are evident in terms of the relative amounts of DNA in both plants regenerated in colchicine (double pattern) and in non-regenerable calli (triple and quadruple patterns). However, the samples of the plants analyzed that exhibited double patterns did not present an intense green color, showing lethal morphological deviations that caused them not to adapt to greenhouse conditions. As expected, the samples from control plants (regenerated without colchicine) showed bright green leaves and evidenced simple DNA patterns. It is important to highlight that from the point of view of population genetics studies, our results indicate the probability of an increase/modification of nuclear DNA; therefore, the occurrence of phenotypic variability must be verified in the ex vitro adapted clones (genotype x environment).

Although the above results are not straightforward, probably due to the low number of analyzed samples, we followed the strategy of bridging with the traditional selection system for genetic improvement. In this sense, the focus was on mass-producing blueberry clones,

with the premise that the effects of colchicine, whatever they were, including the increase in the appearance of somaclonal variants, would be random [32,33]. Therefore, obtaining clonal populations would increase the probability of selecting the desired phenotypes under real conditions of a genotype–environment interaction. Our view is that the effect of colchicine on the genome of plant species with different degrees of domestication should be studied further by integrating more robust tools, for example, cell sorting [34], qRT-PCR [35], and transcriptomics [36]. However, for genetic improvement purposes, the most important condition is to obtain a wide spectrum of variability that allows for selecting individuals with the desirable combination of phenotypic traits [37].

Using colchicine, the induction of tetraploids in *Vaccinium darrowii* was verified by cytogenetics. In that case, the clones were subsequently crossed with highbush blueberry (*Vaccinium corymbosum*) and the progeny confirmed tetraploids by stomatal guard cell size and pollen size characterization [38]. In the same line, explants of berry (*Vaccinium bracteatum* section *Bracteata*) were treated with colchicine, while the resulting tetraploid individuals were crossed with highbush blueberry 'Spartan' (*Vaccinium corymbosum* section *Cyanococcus*). The authors recommend the resulting intersectional hybrids as parents for their agro-productive characteristics, such as high sugar and phytochemical contents, as well as adaptability to abiotic stress [39]. Additionally, a population of intersectional hybrids was obtained by crossing tetraploid highbush blueberry cultivars with pollen from *Vaccinium staminaum* plants that were characterized as tetraploid after treatment with colchicine. Likewise, promising parents for productive traits are characterized from the resulting F1 population [40].

On the basis of a domesticated donor, a study was published in which blueberry plants (*V. corymbosum*) from the donor M5 were regenerated through organogenesis under poly-ploidization treatments with colchicine and oryzalin. While colchicine had the best results, in that case, the presence of 15 tetraploid individuals and 34 mixoploid individuals was confirmed. In this case, the study of phenotypic variability was limited to a minor number of in vitro plants, while the authors postulate the potential of these clones as progenitors for a breeding program [38]. On the other hand, the role of somaclonal variations in the in vitro propagation of berry plants has been reviewed, providing evidence on both their genetic and epigenetic origins, as well as how genetic regulation is related to the level of DNA methylation. In this sense, the importance of exploring the epigenome of berry plants in different environments is stated to verify whether an epigenetic fingerprint remains in the regenerated plants [39].

After two years of transplanting and adaptation to greenhouses, the presence of phenotypic variability accompanied by flowering (agro-productive trait) was verified in 157 blueberry clones (Table 2), estimating a selection efficiency of 8% up to that stage. Considering the variability in agrobotanical traits and flowering as a productive character, the results demonstrate an increase in the selection efficiency in clones from treatments with greater exposure to colchicine. However, it should be noted that the increase in the selection efficiency of promising clones was not linear with the increase in the exposure time to this antimitotic agent. From this population, it has been possible to characterize reference genes in Vaccinium for further gene expression analysis [26].

In the later stage and after three additional years, 38 blueberry clones were selected from the 157 clones from greenhouses that were transplanted to the field. At this point, our strategy was to increase the selection pressure by simulating adverse environmental conditions based on the following assumptions:

a.　The donor Duke is a genotype introduced and adapted to Chile based on intense agronomic management practices. Therefore, the conditions of minimum tillage and zero agronomic management constitute an environmental pressure for the expression of their wild genetic potential, being the comparative basis for the selection of improved clones.

b.　The stressful selection pressures, which, in this case, correspond to the natural environmental conditions that occurred during the three-year period, allow for

the selection of genotypes with greater adaptability and plasticity, suitable for the sustainable management of this crop.

From the 38 clones with the greatest environmental adaptability and through multivariate analysis of principal components (PCA), it was possible to verify the improvement in the variable fruit size, which is related to agronomic yield. In this case, the grouping of clones at defined points of gravity with contrasting yields was demonstrated, which can be assumed because of the effect of colchicine during the process of organogenesis and plants' regeneration [9,10]. Finally, six elite blueberry clones were selected for their improvement in the performance variables and environmental adaptability studied.

Together, the results validate the initial strategy of carrying out population studies under field conditions that allow for selecting elite individuals because of genetic improvement. The methodology of this work was implemented in a single donor genotype; however, considering the versatility of the regeneration and multiplication system in commercial blueberry cultivars [6,24,31,41], it is possible to predict that it can be extrapolated to a wide range of Vaccinium genotypes. To our knowledge, this is the first report in which blueberry plants of a commercial donor cultivar were regenerated in colchicine, maintaining phenotypic variability (five years) under ex vitro conditions, allowing for the selection of elite clones improved for fruit yield and adaptability to environmental stress conditions.

## 5. Conclusions

An efficient protocol was standardized for regeneration through organogenesis of blueberry plants (*V. corymbosum* L.) in culture medium supplemented with 1 mg/L colchicine. Nodal segments of the Duke donor genotype were exposed to colchicine treatments for 1, 2, 3, 5, and 30 days in WP basal medium with 1 mg/L 2iP. The regenerated plants showed in vitro phenotypic variability for traits related to vigor. A population study of the phenotypic variability was carried out for the first stage (two years in the greenhouse) and second stage (3 years in the field). The maintenance of the phenotypic variability under the experimental conditions studied was significant as a source for the selection of elite individuals improved in their adaptability to environmental conditions, as well as agricultural performance variables associated with larger-sized blueberry fruits compared to the donor genotype Duke. A suitable platform was established for the genetic improvement of blueberries assisted by biotechnology in Chile.

**Author Contributions:** Conceptualization, A.D.A. and V.D.; methodology, R.H., A.L. and A.G.; formal analysis, R.H., A.L., B.V. and A.G.; investigation, R.H., A.L., B.V. and A.G.; data curation, R.H. and A.L.; writing—review and editing, R.H., A.L. and A.D.A.; supervision, V.D.; project administration, A.D.A. All authors have read and agreed to the published version of the manuscript.

**Funding:** This work received financial support from the Regional Government of Maule through the projects entitled: Recommendation of new varieties of berries obtained through biotechnology, FIC Maule BIP 40.001.114-0, and BIP N° 30.386.978-0.

**Conflicts of Interest:** The authors have no conflicts of interest to declare.

## Abbreviations

BAP-6—Benzylaminopurine; 2iP-N6—[2-isopentenyl] adenine.

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
