# Peer review of "Organogenesis of Plant Tissues in Colchicine Allows Selecting in Field Trial Blueberry (Vaccinium spp. cv Duke) Clones with Commercial Potential"

_horticulturae, doi:10.3390/horticulturae10030283_

Round 1
Reviewer 1 Report
Comments and Suggestions for Authors
The article deals with an important and actual topi: the genetic improvement of blueberry. The small fruits consumption and cultivation, blueberry in particular, is increasing worldwide. So it is crucial to find new cultivars adapted to non-traditional cultivation areas with high yield potential and fruit quality. The manuscript it is well written and the experiment was conducted rigorously. I have only some minor comments:
-in the abstract I suggest to introduce a sentence to introduce the topic and after the aim of the study
-keywords: avoid words already present in the title
- units of measurement carefully check and uniform to journal style
-l. 52-55 better contextualize or eliminate this part
-5 min of immersion in ethanol it seems a lot, please carefully check
-l. 118 b-trinodal is bi-trinodal
- I suggest to replace seedlings with proliferated explants/shoots
-table 1 add standard error and ANOVA if applicable
-table 3 use as a decimal separator . and not ,
- avoid the use of personal form (in our study/our view)
Author Response
-Dear reviewer, thanking you for your critical comments and review of the manuscript, please see the modifications made to the document indicated in yellow:
-in the summary I suggest introducing a sentence to introduce the topic and then the objective of the study (done)
-keywords: avoid words already present in the title (repeated words eliminated)
- units of measurement carefully review and standardize to the magazine style (done)
-l. 52-55 better contextualize or remove this part (corrected, part removed)
-5 min immersion in ethanol seems like a lot, please check carefully (added reference of our previous work https://doi.org/10.3390/plants11192645 sect. 4.1)
-l. 118 b-trinodal is bi-trinodal (b refers to the numbering of the statement not to the tissue segment)
- I suggest replacing seedlings with proliferated explants/sprouts (corrected, replaced in 3 of 4 parts where “seedlings” appears based on the context of the paragraph)
-table 1 add standard error and ANOVA if applicable (the data type does not apply for the anova type analysis, however it was applied for table 3)
-table 3 use as decimal separator. and no, (corrected)
- avoid using the personal form (in our study/our opinion) (done)
Reviewer 2 Report
Comments and Suggestions for Authors
The manuscript presents a methodology for the regeneration and selection of elite blueberry clones through a five-year experimentation. Vacinium corymposum is a very important fruit of grate economical value.
Hence the selection of new varieties is a challenge for the horticulture science.
However, I have some suggestions that should be addressed before publication.
I think Table 1 and 3 should include one-way ANOVA analysis.
TITLE:
l. 3/define the species (Vaccinium corymbosum L. cv Duke)
ABSTRACT
l. 19/use uniform units (check everywhere in the text)
l. 19/write the full name of the 2iP and BAP, and then write the abbreviations.
l. 25/ revise ‘6 787’
KEYWORDS
Delete words that are included in the title
INTRODUCTION
l. 54/ delete our country / replace with “Chile”
MM
l. 106/“Adventitious buds (~5 cm)”? / Something is wrong here? do you mean stem (~5 cm)?
l. 107/ did you follow the presented disinfection protocol? First NaOCl, then ethanol following by rinsed in sterile water?
l. 109/ 1-2 references are needed (to explain the use of WPM and 1mg/L 2iP)
l. 113,136,137, 187/ revise units thoroughly in the text
l. 144 / revise to (2:1, v/v)
l. 148, 161 / you have to explain more how did you evaluate these traits / write information please / are they quantitative or qualitative characteristics? You could refer a relative manuscript
l. 159/ how did you count brix?/give information please (method, equipment…)
RESULTS
l. 187,190/revise units
Table 1,3/statistical analysis is missing. It is necessary.
Table 1/it is not clear/what do you count?
Table 1,3/revise units please
Table 2 / what do you mean multiplication?
Table 3 / explain sympbol “Δ’’
l. 271/ revise data
l. 380 / revise MCPA to PCA (you don’ t use MCPA again)
DISCUSSION
l. 508-509/ I think you have included these in Conclusions, It is very important for your research
CONCLUSIONS
l. 524-5/ please delete ‘L (N6-[2-isopentenyl] adenine)’ and ‘McCown's Woody Plant’
l. 528-531 /please re-write: it is not clear
REFERENCES
It needs to be revised thoroughly; the format has many issues
Author Response
-Dear reviewer, thanking you for your critical comments and review of the manuscript, please see the modifications made to the document indicated in yellow:
I think Tables 1 and 3 should include a one-way ANOVA analysis. ((this suggestion was made for table 3 with a Kruskal Wallis analysis (non-parametric anova))
QUALIFICATION:
l. 3/define the species (Vaccinium corymbosum L. cv Duke) (corrected)
SUMMARY
l. 19/use uniform units (check everywhere in the text) (corrected, mg/L units were fixed for mg/l)
l. 19/write the full name of the 2iP and BAP, and then write the abbreviations. (added in abbreviations)
l. 25/ review "6,787 (6,787 refers to the number of floors, corrected)
KEYWORDS
Delete words included in title (corrected)
INTRODUCTION
l. 54/ delete our country / replace with "Chile (corrected)
MM
l. 106/ "Adventitious buds (~5 cm)"? / Something is wrong here? Do you mean stem (~5 cm)? (corrected)
l. 107/ Have you followed the disinfection protocol presented? First NaOCl, then ethanol, then rinse in sterile water? (modified order, ethanol first then NaOCl)
l. 109/ 1-2 references needed (to explain use of WPM and 1mg/L 2iP) (Corrected, two references added using 2ip and WP for Vaccinium callus induction in WPM)
l. 113,136,137, 187/ check the units in the text carefully (done)
l. 144/ revise to (2:1, v/v) (corrected)
l. 148, 161 / you have to explain more how you have evaluated these traits / write information please / are they quantitative or qualitative traits? You could refer to a relative manuscript (done!!) (I added information about whether the traits are qualitative or quantitative and I also added the citations of two articles that made measurements similar to ours)
l. 159/ How did you count the brix? /give information please (method, equipment...) (added)
RESULTS
l. 187,190/review units (corrected)
Table 1.3/statistical analysis is missing. It is necessary. (Table 1 does not correspond to the type of data, it is only a summary of experiments,) (statistical analysis was applied to Table 3, along with this, part of the statistical analysis methodology was modified)
Table 1/unclear/what counts? (Summary of regeneration rate of blueberries in different concentrations of colchicine in leaf discs and nodal segments isolated from Duke vitroplants.)
Table 1.3/check units please (corrected)
Table 2 / what does multiplication mean? (refers to the multiplication factor of explants in colchicine)
Table 3 / explain symbol "Δ'' explained!)
l. 271/ review data (corrected, added a comma for separation)
l. 380 / review MCPA to PCA (you do not use MCPA again) (388 / it is not another PCA analysis, it is a clustering graph according to the k means distribution of the principal components identified in PCA that allows differentiating two groups based on of agricultural performance variables)
DISCUSSION
l. 508-509/ I think you should include this in Conclusions, it is very important for your research. (I am in a different place due to modifications to the text)
CONCLUSIONS
l. 524-5/ please delete 'L (N6-[2-isopentenyl] adenine)' and 'McCown's woody plant' (corrected)
l. 528-531 /please rewrite it: it is not clear (this refers to the author contributions)
REFERENCES
Needs a thorough overhaul; the format has many problems (corrected)
Round 2
Reviewer 2 Report
Comments and Suggestions for Authors
Perhaps it was not clear. The correct units are mg/L.
Please, fix it again.
Author Response
Dear reviewer.
Please see the manuscript amended by minor revision as suggested
Thank you
Prof. Ariel D. Arencibia Ph.D